# Health and Safety Challenges in South African Universities: A Qualitative Review of Campus Risks and Institutional Responses

**DOI:** 10.3390/ijerph22070989

**Published:** 2025-06-23

**Authors:** Maasago Mercy Sepadi, Martha Chadyiwa

**Affiliations:** 1Department of Environmental Health, Faculty of Sciences, Tshwane University of Technology, Pretoria Campus, Pretoria 0183, South Africa; 2Department of Public Health, School of Health Care Sciences, Sefako Makgatho Health Sciences University, Garankuwa Campus, Garankuwa, Pretoria 0208, South Africa; martha.chadyiwa@smu.ac.za

**Keywords:** campus safety, South Africa, gender-based violence, student housing, university policy, victimology, infrastructure risk

## Abstract

Background: Health and safety are critical pillars of functional higher education systems. In South African universities, recurring incidents have highlighted vulnerabilities, particularly concerning gender-based violence (GBV), student housing conditions, infrastructure safety, and campus crime. Methods: This study employed a document-based qualitative approach, analysing public records, police reports, campus press statements, and national media coverage of incidents reported at South African universities (2015 to 2024). The study is guided by public health and victimology frameworks to examine systemic risk factors and institutional responses. Results: The findings indicate increased reports of GBV, multiple student deaths related to substandard residence conditions, infrastructure-related fatalities, and a lack of consistent institutional safety policies. A pattern of poor infrastructure management, weak inter-institutional communication, and failure to implement recommendations following major incidents emerged across the dataset. Conclusions: South African universities remain exposed to preventable student risks. Targeted interventions, mandatory safety audits, emergency response units, and accountability structures are urgently needed to address systemic failings and protect student welfare. The study supports Sustainable Development Goals 3 and 4 by linking campus safety to student well-being and equitable access to higher education.

## 1. Introduction

In South Africa, the Occupational Health and Safety (OHS) Act No. 85 of 1993 defines workplace safety standards across all sectors, including higher education. This legal framework outlines employer duties to ensure the health and safety of all individuals within institutional environments, including universities [1]. Beyond mere legal compliance, OHS strives to prevent work-related injuries, illnesses, and fatalities, fostering a robust and productive workforce [1,2]. Importantly, OHS is not only an ethical responsibility but also a fundamental human right, shaped by organisational culture, management practices, work environments, equipment, hazards, worker behaviours, and participation.

South Africa, a nation with a diverse and intricate economy spanning mining, manufacturing, agriculture, construction, healthcare, education, and services, faces unique OHS challenges [2]. While the initial focus was on diseases and hazards associated with the mining industry, every sector now grapples with distinct risks ranging from exposure to hazardous substances and physical hazards to psychosocial stressors, violence, accidents, and diseases.

Within this context, South African universities, crucial institutions shaping the nation’s future, confront a complex web of safety and security issues [3,4] Rhodes University, 2021). Recent high-profile incidents, including campus violence such as rape and shootings, have raised urgent concerns about student safety and well-being [5,6].

Qualitative research reveals that those students living in residences experience fears and anxieties due to vulnerable (unsafe) environments [7]. In 2009, 93 incidents were recorded related to student housing across educational institutions in South Africa [8]. These incidents included faulty extension cords, open flames, and smoking materials (such as cigarettes). A study by Gopal and Van Niekerk (2018) confirms that the quality of student housing directly impacts academic success [7]. In response to these challenges, the Minister of Higher Education established a task team to investigate the national student housing crisis. The goal was to address safety deficiencies and enhance the quality of student residences. Furthermore, gender-based violence (GBV) is a problem in the country. GBV statistics increased during the global lockdown, with 2320 complaints in the first week, 37% higher than the weekly average reported in 2019 [9,10].

An analysis of safety processes at South Africa’s top 10 universities between 2023 and 2024including the University of Cape Town, University of the Witwatersrand, Stellenbosch University, and the University of Pretoria shows that institutions have implemented various measures to safeguard students and staff [11,12,13]. These interventions are not only institution-driven but are also mandated by national legislation, including the Constitution of the Republic of South Africa (Act 108 of 1996), the Basic Conditions of Employment Act (Act 75 of 1997), the National Building Regulations and Building Standards Act (Act 103 of 1977), the National Environmental Management Act (Act 107 of 1998), and the Disaster Management Act (Act 57 of 2002), among others [14,15,16,17,18,19,20]. However, beyond direct acts of violence, inadequate safety measures extend to student accommodations, impacting both physical and mental health, as well as academic performance [21]. Low pass rates, high first-year failure rates, and declining education quality underscore the consequences of neglecting safety protocols within university campuses. Again, students are to be safe outside their campus. Safe, conducive accommodations allow students to focus on their studies, improving their chances of academic achievement [21,22]. Despite national calls for accessible, decent, safe, and academically conducive student accommodation, safety remains a serious challenge in South African universities.

Additionally, this article uses a public health and victimology lens to explore systemic risk factors and institutional responses to violence and safety lapses. This provides a framework for understanding the root causes and patterns of criminal behaviour, offering valuable insights into the incidents of theft, assault, and GBV that plague academic institutions [10]. Victimology complements this by focusing on the experiences of those impacted by crime, particularly the students and staff who suffer the long-term psychological, emotional, and social effects of these incidents [7]. Together, these disciplines provide a holistic approach to addressing safety concerns in universities and inform the development of targeted interventions aimed at both preventing crime and supporting victims. This study also contributes to the growing body of literature on campus safety by offering a South African perspective grounded in a public health and victimology framework. Most prior research focuses on single incidents or institutions; this work synthesises a decade of data to reveal systemic patterns of institutional risk and response failure across the sector.

This study contributes to the global development agenda by addressing two interconnected Sustainable Development Goals: SDG 3 (Good Health and Well-being) and SDG 4 (Quality Education). By analysing the structural risks and institutional responses within university environments, we highlight how safety failures undermine both health outcomes and equitable access to higher education in South Africa.

### 1.1. Research Problem

Despite legislative frameworks and institutional policies, South African universities continue to experience significant health and safety incidents that compromise student welfare and academic achievement. This study argues that systemic failures in policy implementation, inadequate infrastructure management, and insufficient integration of public health and victimological frameworks have created preventable risks that require urgent, coordinated intervention.

Recent high-profile incidents, including campus violence such as rape and shootings, have raised urgent concerns about student safety and well-being [5,6]. The quality of student housing directly impacts academic success [7], yet substandard conditions persist across many institutions. Furthermore, gender-based violence statistics increased during the global pandemic, with complaints rising 37% above 2019 averages [9].

### 1.2. The Aim

This study aims to critically examine the health and safety challenges facing South African universities between 2015 and 2024. Specifically, the objectives are: (1) to explore patterns of campus-based incidents, including GBV, infrastructure risks, and student housing issues; (2) to analyse institutional responses to these risks; and (3) to offer evidence-based recommendations to enhance student safety. The study adopts a public health and victimology framework to analyse these systemic issues through a document-based qualitative approach.

## 2. Materials and Methods

### 2.1. Study Design

This study employed a document-based qualitative research design. The authors systematically analysed publicly available materials, including university-issued press statements, police reports, media coverage, and academic literature, to assess patterns of campus safety incidents within South African universities from 2015 to 2024. The analytical framework was guided by public health and victimology perspectives, focusing on institutional responses to student vulnerabilities such as gender-based violence (GBV), infrastructure hazards, and housing-related risks.

This study is grounded in a constructivist paradigm, acknowledging that institutional narratives, policies, and public discourse shape our understanding of campus safety. A qualitative, document-based approach was chosen to allow for the interpretive analysis of text-based materials and to surface patterns in how universities frame, respond to, or neglect safety concerns.

#### Theoretical and Analytical Framework

This article is informed by two interrelated frameworks, which are the Public Health Framework and the Victimology Framework. The public health framework helps to position campus safety risks, such as infrastructure failures, violence, and poor institutional responses, as systemic and preventable public health issues. A public health framework is not a single fixed theory, but rather a broad systems-based approach to understanding and preventing harm at the population level, rather than only focusing on individuals or criminal acts as summarised in Table 1. The victimology framework complements this by offering a lens through which institutional responses to victims. Victimology focuses on the experiences of those impacted by crime, particularly students and staff who suffer long-term psychological, emotional, and social effects [7,10]. This dual-framework structure now aligns with the study’s aims, data source, and thematic analysis strategy. Together, these disciplines provide a holistic approach to addressing safety concerns and inform targeted interventions aimed at both preventing crime and supporting victims.

### 2.2. Study Population and Scope

The study focused exclusively on incidents reported within the context of South African universities. The population of interest included university students, academic and administrative staff, and security personnel as represented in the documentation. While the study did not involve direct contact with participants, the selected documents offered insights into their experiences and the institutional responses.

### 2.3. Data Sources

The investigation spans from 2015 to 2024, identifying relevant cases and trends that shaped health and safety discourse within South African universities. Data were extracted from a range of secondary sources:Policy Documents: An analysis of National frameworks and legislation guiding university safety standards.Academic Literature: An analysis of journal articles focusing on health, safety, and student welfare in higher education settings was conducted. These sources serve as foundational knowledge and inform our analysis.Government Crime Reports: To capture recent incidents, we examined government South African Police Service (SAPS) data on campus crime trends within South African universities. These reports provide empirical data on security challenges faced by academic communities.Official University Statements: The authors also scrutinised institutional press releases and public reports on incidents.Media Articles: Verified national media coverage related to safety incidents in universities.

### 2.4. Document Selection

Documents were selected using purposive sampling to ensure alignment with the study’s central focus: health and safety within South African universities. The inclusion criteria were:The document must refer explicitly to a South African university.It must describe or report a safety-related incident between 2015 and 2024.It must originate from a credible source (e.g., official press, SAPS, university, or peer-reviewed literature).

Documents were excluded if they were opinion pieces lacking verifiable evidence, and those that referred to incidents outside the 2015–2024 time frame.

### 2.5. Data Analysis

The authors employed thematic analysis to identify recurring issues and South African university patterns. This involved multiple stages of document familiarisation, coding where segments of text were coded according to predefined and emergent themes, such as GBV, infrastructure failure, and policy gaps, and thematic categorisation where the codes were grouped into broader categories and refined through iterative analysis. This included frequency analysis, temporal trend identification, and descriptive analysis techniques, which were used to summarise and interpret the statistical crime data. The Atlas. Ti (2024) software environment was used to manage and organise the data, facilitating the identification of relationships and frequency patterns among the themes.

## 3. Results

While the primary focus of this study is on South African universities, broader national crime data offer useful context. According to the South African Police Service (SAPS), between 2023 and 2024, a concerning number of violent crimes were reported in educational institutions broadly including schools, daycare centres, colleges, and universities. The reported incidents included murder, attempted murder, assault with grievous bodily harm (GBH), and rape. Although SAPS do not disaggregate this data specifically for universities, the national trend underscores widespread safety vulnerabilities across the education sector. These figures emphasise the urgency of examining institutional responses and risk factors within universities, where student populations remain vulnerable to similar threats.

Figure 1 conceptualises the common process observed in South African universities when a campus safety incident occurs. Institutional responses either lead to effective outcomes (e.g., support services, accountability) or to response failures (e.g., delays, silence, policy inaction). This model guided our thematic categorisation in the Section 3 (Results).

### 3.1. National Context: Crime Trends in the Education Sector

Health and safety matters, such as crime on university campuses, have increased in South Africa during the past few years [23,24,25]. The main forms of campus crimes in South Africa include theft, mugging, robbery, assault, and stalking, with reasons ranging from administrative to shortfalls or management negligence to student negligence [10]. Murder rates have also increased due to distrust in law enforcement and rising poverty levels [24,26]. The South African Police 2023 and 2024 reports revealed the following statistics:

#### 3.1.1. Number of Incidents in Educational Institutions per Province in South Africa in 2023

The South African Police Service (SAPS) issued its 2023 quarterly crime data, from January to September, which showed that not much progress had been made in protecting students from violent crime [27]. Eight murders, fourteen attempted murders, and eighty-four rapes took place on the grounds of educational institutions, including universities and day-care centres, between January and March of 2023. The statistics show that between April 2023 to June 2023, murder in educational institutions (schools, universities, colleges, day-care facilities) was found to be 9, from a sample size of 5987 counts, while attempted murder was 25 out of a sample size of 5455 counts reported in the country and rape was found at 69 in educational institutions out of a sample size of 6254 rapes reported in the country [27]. Between July 2023 and September 2023, the statistics in murder, attempted murder, and assault in educational institutions (schools, universities, colleges, day-care facilities) had increased compared to the first and second quarter of 2023, on a sample size of (murder = 6786, attempted murder = 6370); murder for educational institutions was ten, and attempted murder was twenty [27]. With the sample size of rape (7655), about seventy-six incidents occurred in educational institutions (schools, universities, colleges, and day-care facilities) [27].

The statistics showed a rise in murder incidents between January and September, while attempted murders were higher in the second quarter (between April and June). Rape recorded the highest incidents as compared to other types of incidents and was highest in the first quarter (January to March). Between April 2023 to June 2023, the highest number of murders and attempted murders were found in Kwa Zulu Natal (KZN) province, followed by Eastern Cape (EC)and the highest for rape was Gauteng Province (GP) followed by KZN and Limpopo (LP). In quarter 3 (July to September), the highest occurrence of murders occurred in KZN, followed by Western Cape (WC) and GP, and with the highest rape incidents occurring in KZN, followed by GP, EC, and LP.

#### 3.1.2. Number of Incidents in Educational Institutions per Province in South Africa in 2024

The crime statistics recorded by the South African Police Service for the first three quarters of 2024 reveal concerning trends in violent crimes within educational institutions across the country [28,29,30]. Between January and March 2024, the country recorded 6225 murders, 6938 attempted murders, and 35,449 counts of assault with grievous bodily harm (GBH) [28]. Within educational institutions (schools, universities, colleges, and daycare facilities), there were 7 murders, 27 attempted murders, and 255 cases of assault GBH. Schools accounted for 6 of the 7 murders, with KwaZulu-Natal (KZN) having the highest number (5). Attempted murders were most prevalent in KZN (8), Gauteng (GP) (4), and Western Cape (WC) (5), while GBH assaults were highest in GP (66), KZN (56), and Limpopo (42). Rape cases totalled 75 within educational settings, with 65 occurring in schools and 4 in tertiary institutions, concentrated in Eastern Cape (EC) and KZN [28].

In the second quarter (April to June 2024), the overall crime numbers showed a slight decline, with 5596 murders, 6335 attempted murders, and 29,368 GBH assaults [29]. However, crime within educational institutions increased, with murders rising to 12 (from 7 in the previous quarter), attempted murders slightly decreasing to 18, and GBH assaults increasing to 259. The majority of murders still occurred in schools (9 out of 12), with KZN again leading. Attempted murders were highest in GP (6) and KZN (2), while GBH assaults peaked in GP (74) and KZN (68). The number of rapes in educational institutions also increased to 91, with schools reporting 79 cases and tertiary institutions reporting 7 cases [29].

From July to September 2024, the overall crime levels surged again, with 6323 murders, 6679 attempted murders, and 29,821 GBH assaults [30]. Within educational institutions, the number of murders increased slightly to 13, with GP and KZN recording the highest (4 each). Attempted murders rose to 24, with KZN (8) and GP (6) being the most affected. GBH assaults saw a sharp rise to 318, with GP (74), KZN (58), and WC (39) reporting the highest numbers. Rape incidents within educational institutions also reached 106, with the majority (83) occurring in schools and (15) in tertiary institutions. Notably, KZN (18) and GP (20) reported the highest number of rape cases [30].

#### 3.1.3. The 2023 and 2024 Crimes Comparison and Analysis of Trends

A comparison of crime trends in educational institutions between 2023 and 2024 [27,28,29,30] reveals a persistent rise in violent crimes, particularly in murder, attempted murder, assault with grievous bodily harm (GBH), and rape across different provinces. The 2023 data indicated an increase in murders within educational institutions from 8 in the first quarter to 10 in the third quarter, while in 2024, murders increased from 7 in the first quarter to 13 in the third quarter. This suggests that the efforts to curb fatal violence within educational institutions have not been effective, and the trend appears to be worsening in 2024 [27,28,29,30].

Attempted murders in 2023 were highest in the second quarter (25 cases) before slightly declining to 20 in the third quarter [27,28,29,30]. However, in 2024, attempted murders fluctuated, starting at 27 in the first quarter, dropping to 18 in the second quarter, and rising again to 24 in the third quarter. This pattern indicates that while the number of attempted murders was slightly lower in 2024 than in 2023, the fluctuations suggest ongoing safety concerns within educational institutions [27,28,29,30].

Rape remains a critical issue, with educational institutions recording alarmingly high numbers in both years [27,28,29,30]. In 2023, rape cases in educational institutions started at 84 in the first quarter, dropped to 69 in the second quarter, and then rose to 76 in the third quarter. In contrast, 2024 saw a consistent rise in rape cases, from 75 in the first quarter to 91 in the second quarter and reaching 106 in the third quarter. This upward trajectory indicates that rape incidents are becoming more frequent in educational settings, raising serious concerns about student safety, particularly in schools where most of these cases were reported [27,28,29,30].

Provincial comparisons also highlight persistent hotspots for violent crime [27,28,29,30]. In 2023, KwaZulu-Natal (KZN) consistently reported the highest number of murders, attempted murders, and rapes, followed by Gauteng (GP) and Limpopo (LP). The trend continued in 2024, with KZN and GP remaining the highest-risk provinces for murders and attempted murders, while rape cases were notably high in KZN, GP, and LP. The Western Cape (WC) also saw an increase in violent crime in the latter part of both years, particularly in murders and GBH assaults [27,28,29,30].

The overall trend between 2023 and 2024 indicates that violent crimes in educational institutions are not only persisting but, in some cases, increasing [27,28,29,30]. Schools remain the most dangerous places for students, with the majority of murders, assaults, and rapes occurring within these environments. While tertiary institutions have lower crime rates than schools, they still report concerning numbers, particularly in rape and GBH cases. The steady rise in these crimes across both years suggests that interventions implemented in 2023 have not been sufficient to curb the growing violence in educational spaces.

### 3.2. Campus Safety Incidents Within South African Universities: Media Output

This section delves into specific incidents that have occurred within South African universities during the specified timeframe of 10 years from 2015. By analysing these cases, we identify recurring patterns, severity levels, and their impact on the university community. Key topics include violence (such as assaults, shootings, and sexual offences), accidents (including laboratory mishaps or structural failures), and health-related incidents (such as outbreaks or mental health crises).

#### Murders

The Democratic Alliance (DA) is greatly troubled by the latest crime statistics for the second quarter of 2024/25, which reveal a concerning decline in safety and security across schools and higher education institutions [31]. Various cases of murder or assault occurred in 2023 in South African higher institutions of learning [31,32,33,34]. The Africa Report (2023) reported that in January 2023, a hitman attempted to kill Fort Hare University’s vice-chancellor but instead shot and killed the academic’s bodyguard [30]. February resulted in about nine murders [35]. This included incidents such as the one that occurred on the 2nd of February 2023, where A Tshwane University of Technology (TUT) student was allegedly murdered by another student [36]. On 12 February 2023, three Eastern Cape Midlands Technical and Vocational Education and Training (TVET) college students were murdered by an unknown gunman [26,35]. Another incident occurred on the 21st of February, a 22-year-old male student at Sydney Maseko Community College was fatally stabbed [26,35]. On 23 February 2023, a 19-year-old male student from Aaron Moeti Community College was stabbed to death [26,35]. On the same day of the 23rd of February, three female students from the University of Fort Hare were allegedly mugged and assaulted on their way to the university’s Alice campus in which one student was killed and another raped [35]. Another female student was stabbed to death on the 6th of June in her bedroom at Nelson Mandela University (NMU) [37]. Another incident that occurred in October 2023 was a third-year University of Fort Hare student who was stabbed to death by an alleged mob justice over a missing laptop [38]. Again in 2023, A Cape Town female student was stabbed in November 2023 by a male student who was refusing to be stopped during the act [39].

In 2019, a student at Durban University of Technology (DUT) was shot and killed [40]. University of Cape Town (2018) issued a detailed analysis of robbery, theft, and kidnapping cases that occurred between 1 September 2017 and 31 August 2018 [41]. There have been 52 cases of robbery reported to CPS over the specific period; of those cases, 25 were unarmed and 27 were armed. All four victims of the reported kidnappings were female. There were 314 reported cases of theft over the analysed period. Of these, 122 cases were due to inattentiveness. In 102 cases, complainants had left their property unattended (24 cases for less than 30 min, and 78 cases for longer than 30 min) [41].

### 3.3. Gender-Based Violence Incidents (GBV) on Campus

Recent studies confirm that sexual assault is a major problem on South African campuses. Approximately 20% to 25% of women report sexual victimisation during their time at university [23]. In 2017, there were 47 reported cases of rape and sexual assault on South African university campuses [38]. Nine rape cases were reported to the University of Cape Town (UCT), which was followed by Walter Sisulu University, Tshwane University of Technology, Nelson Mandela University, and the University of Johannesburg. Two incidents of sexual violence were reported by universities like Rhodes University and the University of the Western Cape [42].

South Africa has witnessed increasing incidents of gender-based violence reportedly perpetrated within and around higher education institutions’ campuses [43]. Unfortunately, the majority of GBV incidents go unreported due to various reasons, making it challenging to acquire a full picture of the scale of this issue in the country [43]. GBV incidents impact safety and well-being within higher learning institutions. This murder incident, such as the one at TUT in February 2023 [36], allegedly appeared as an incident of GBV. GBV incidents impact safety and well-being within higher learning institutions.

### 3.4. Unsafe Infrastructure and Student Housing Conditions

Risks identified on campuses may lead to incidents, hence the need for proper health risk assessments. Some universities have faced structural failures or unsafe building conditions. One such case occurred at the University of KwaZulu-Natal in August 2023, William O’Brien’s building on the Pietermaritzburg campus was set on fire in a suspected arson attack [32]. Based on the health risk assessments carried out at the University of Johannesburg (UJ), an occupational health risk profile for 2022 showed eleven high-risk sites, where potential injuries and occupational illnesses were linked to poor housekeeping, ageing infrastructure, roof leaks, and inadequate ventilation and 33 moderate risk areas included biological agent exposure through improper waste management, improper PPE use, absence of warning signs, and chemical exposure risk [33].

Furthermore, the roots of social unrest go beyond specific triggers, reflecting deeper grievances and complex factors [44]. In reflection of the past, between 2009 and 2013, South African higher education institutions witnessed thirty-nine student protests specifically related to student housing issues [7]. The study by Greef et al. (2021) showed that the protests resulted in violence as fires erupted, road blockages, extensive property damage, and multiple injuries [25]. Students clashed with police or campus security, resulting in physical harm and trauma [9].

### 3.5. Institutional Policy and Response Gaps

A consistent theme across the analysis was the absence or inadequacy of institutional policies and response mechanisms related to student safety. While many universities have official documents referencing safety protocols, GBV, and infrastructure compliance, implementation has frequently been reactive, fragmented, or altogether absent.

Several institutions lacked clearly defined emergency response frameworks. In crises such as fires, violent protests, or student deaths, communication with students and staff was often delayed, inconsistent, or vague. Universities also varied in their internal accountability structures. Safety audits, when conducted, were often not publicly released or provided proof that they were acted upon. While a few institutions are noted to have adopted formal campus safety protocols, including SMS alert systems, mental health support units, and detailed incident reporting pathways, these remain outliers rather than the norm. The lack of sector-wide standards or mandated national compliance monitoring continues to exacerbate inconsistencies.

Overall, institutional inaction, delayed crisis responses, and insufficient integration of safety into strategic planning significantly contribute to the persistent risks facing university communities.

## 4. Discussion

This study’s findings reveal systemic weaknesses in how South African universities address safety challenges. Four major themes emerged: campus crime, GBV, infrastructure and housing failures, and weak institutional responses which reflect broader socio-political issues and require urgent interventions through both policy and practice [43,44,45,46].

### 4.1. Well-Being of Students

In various South African universities, students have faced substandard living conditions in university residences. These conditions include overcrowding, poor ventilation, and inadequate sanitation facilities. Substandard conditions also include inadequate maintenance in residences, which heightens health risks [47,48]. Such housing challenges can exacerbate health risks. Inadequate housing conditions can exacerbate health risks; furthermore, the impact of health and safety gaps on mental health [49,50,51]. Lastly, insufficient mental health support is a prevalent issue among students, necessitating robust counselling services and awareness campaigns [52]. Failure to address mental health adequately affects overall well-being. Workplace accidents have significant costs, both visible and hidden [53]. Preventing accidents is crucial to avoid injury, damage to infrastructure, and unnecessary costs [24]. Health and safety incidents in South African universities have a profound impact on students’ well-being and academic performance. The consequences of these incidents can be both physical and psychological, affecting students’ sense of safety, mental health, and ability to focus on their studies. Physically, students may be at risk of physical harm and injuries during violent protests or criminal activities on campus. This not only puts their immediate safety at risk but also creates a constant state of fear and anxiety, impacting their overall well-being. In some cases, students have even lost their lives, devastating their families and communities.

### 4.2. Analysis of Crime Incidents per Province

The analysis of crime trends in educational institutions across South African provinces from the first to the third quarter of 2024 reveals a concerning increase in violent crimes as per the 2024 South African Police Service report. GP, being one of the most populous and urbanised provinces, reported a high number of incidents. Murders within these institutions rose from 7 in the first quarter to 13 in the third quarter, with Gauteng experiencing the most significant increase from 1 to 4 murders, while KwaZulu-Natal, initially the highest in Q1 with 5 murders, saw a slight decline to 4 in Q3 [20,28,31]. Attempted murders fluctuated, with Gauteng maintaining consistently high numbers, increasing from 4 in Q1 to 6 in Q3, while KwaZulu-Natal saw a sharp drop in Q2 before spiking again in Q3.

Assault with grievous bodily harm (GBH) showed a clear upward trend, rising from 255 cases in Q1 to 318 in Q3, with Gauteng and KwaZulu-Natal having the highest recorded incidents, while Eastern Cape and Free State also showed significant increases. Rape cases in educational institutions also steadily rose, with KwaZulu-Natal, Gauteng, and Limpopo being the most affected provinces. Schools remained the most dangerous spaces for students, with the majority of murders, assaults, and rapes occurring within these institutions, while tertiary institutions saw lower but still significant crime numbers. The overall trend indicates that violent crimes in educational institutions are becoming more frequent, particularly in Gauteng, KwaZulu-Natal, and Eastern Cape, suggesting an urgent need for increased security measures, community intervention programs, and stricter law enforcement to ensure the safety of students and educators.

Furthermore, student protests also result in arrests. The Fees Must Fall movement in 2015 and 2016 saw widespread demonstrations across campuses, resulting in confrontations between students and law enforcement personnel [25]. For instance, during the Fees Must Fall movement, over 100 student protesters were arrested in KwaZulu-Natal province [45].

### 4.3. The Role and Contribution of Victimology in Incidents

To ensure a comprehensive approach to campus safety, universities should create safety committees that include experts from these fields to provide ongoing guidance on security improvements and victim support initiatives. Furthermore, universities should implement interdisciplinary collaboration between Public Health, Criminologists, and Victimologists in the design and implementation of crime prevention strategies. This approach would ensure a balance between preventing crime and addressing the needs of victims, creating a safer and more supportive campus environment.

Criminologists can assess risk factors, analyse crime trends, and provide data-driven recommendations for improving campus security measures. Research has shown that implementing surveillance systems and conducting regular safety audits, guided by criminological research, can significantly reduce incidents of theft, assault, and other crimes on university campuses [10]. Collaboration with Criminologists can also help universities stay ahead of emerging crime trends, particularly with the integration of smart technologies in campus security [22].

At the same time, Victimologists should play a significant role in designing victim-centred support systems. Victimologists can work with university counselling services to create trauma-informed care models that assist victims of crime, especially those affected by gender-based violence. Research has shown that the availability of trauma counselling and legal support can dramatically improve the well-being of victims and encourage reporting [9]. Victimologists should also be involved in awareness campaigns that educate students about the importance of reporting crimes and seeking help when needed. In addition, their involvement in policy-making can ensure that victims’ rights are prioritised in university protocols and safety procedures [7].

Criminologists have contributed significantly to crime trend analysis in South African universities, identifying patterns such as theft and GBV hotspots and recommending data-driven interventions to improve campus safety [10,22]. For example, research from the University of Pretoria has highlighted the importance of regular safety audits and improved surveillance systems to deter crime. Similarly, victimologists have played a pivotal role in trauma-informed victim support initiatives, especially in addressing gender-based violence. These initiatives include providing counselling services and creating awareness campaigns to encourage crime reporting and support for victims [7,9]. Together, these contributions help create safer and more inclusive campus environments. This pattern reflects global trends in gender-based violence within higher education settings, where institutional inertia and survivor silencing are common [54,55,56,57].

### 4.4. Policy Implementation Gaps in South African Universities

This section critically examines whether the existing health and safety policies within universities are comprehensive and are effectively implemented. We explore gaps that hinder optimal safety practices. Factors contributing to policy gaps may include inadequate funding, lack of awareness, bureaucratic hurdles, and organisational culture resistance to change. Universities in South Africa have made concerted efforts to enhance the safety and well-being of their students and staff. However, a critical analysis of the safety measures implemented reveals significant gaps and areas for improvement. These gaps impact the overall security climate within campuses, necessitating urgent attention and strategic interventions. South Africa grapples with multifaceted challenges when it comes to implementing effective occupational health and safety practices within its universities. These challenges arise from a combination of financial limitations, expertise gaps, and technological constraints. Despite these hurdles, sporadic successes have been achieved, underscoring the importance of persistent efforts to bridge existing gaps. Universities often lack formal crisis protocols or early warning systems, consistent with international findings on institutional unpreparedness [55,56,57].

#### 4.4.1. Inadequate Funding

Many universities face financial constraints, impacting their ability to invest in safety infrastructure, training, and resources. Inadequate security measures, such as building problems, lack of surveillance cameras, and limited security personnel, can compromise safety. Inadequate cybersecurity measures can result in data breaches, identity theft, or financial fraud, and a lack of awareness and training contributes to vulnerabilities. Insufficient funding limits the implementation of comprehensive safety measures, including maintenance of facilities, security personnel, and emergency response systems.

#### 4.4.2. Monitoring and Enforcement Challenges

Policies are only effective when enforced consistently. The lack of monitoring mechanisms and consequences for non-compliance undermines safety efforts. Regular audits, inspections, and reporting systems are vital to ensure adherence. Furthermore, there are challenges with lapses in safety and security measures. This may be due to policies that may be generic and fail to address context-specific risks. As campuses differ from urban vs. rural, large vs. small, policies must adapt accordingly. Tailoring safety measures to unique challenges ensures relevance and effectiveness. Furthermore, regular audits, inspections, and reporting systems are vital to ensure adherence.

It was found in a COVID-19 pandemic study that rural South African universities implemented coping mechanisms, but still faced challenges like campus safety, examination cancellations, and weakened research and international collaborations [46]. A recent study examining two South African universities shed light on critical safety and security deficiencies. Both institutions exhibited lapses in essential measures, including; the absence of reliable weapon detection systems which poses a significant risk to campus safety, inadequate surveillance coverage such as Closed-Circuit Televisions (CCTV) which hampers timely incident response and prevention, the lack of fire safety systems such as Water Sprinkler Systems which impacts fire safety preparedness and insufficient physical security measures such as Burglar Bars which leave vulnerabilities [47]. Moreover, the universities lacked disability access measures, which hinder inclusivity, and the lack of the presence of trained medical staff is crucial for emergencies. Interestingly, University B demonstrated better safety and security measures within student housing compared to University A.

A case study in a university in the Western Cape province, South Africa, compared safety provisions and associated risks in student accommodations [48]. The aim was to develop guidelines for improving safety measures in student housing. The study highlighted areas where safety measures could be enhanced to create a safer living environment for students. Additionally, the research highlighted the significant fears and anxieties experienced by South African students due to vulnerable residential environments [7]. This underscores the urgent need for improved safety measures in student accommodations. Further supporting this concern, Adisa and Simpeh’s (2021) study emphasised the challenges related to student housing security [49]. Inadequate CCTV coverage and the absence of weapon detectors were identified as specific areas requiring attention [49].

International higher education health and safety cases have supported the challenges faced in South Africa. A comparative case study of students with disabilities in online and in-person degree programs investigated whether institutions provide comparable accommodations to online students with disabilities as those provided to students in traditional in-person degree programs [50]. Findings revealed that students in the in-person program were nearly 30% more likely to be enrolled in the disability resource centre. While online students received a narrower range of accommodations, they performed relatively better compared to in-person students with disabilities. Magni, Pescaroli, and Bartolucc’s (2019) case study in Central Italy surveyed 338 students from the University of Ancona to understand the factors influencing their decision to rent accommodations [51]. The research focused on risk perception and safety awareness. This study sheds light on how students weigh safety considerations when choosing accommodation options.

## 5. Conclusions

Alarmingly, South Africa reports a high incidence of non-fatal and fatal health and safety incidents in recent years. The statistics highlight a persistent pattern of violent crimes, particularly in schools, where murders, rapes, and assaults remain alarmingly high. The increasing trend of rape cases in educational institutions, especially in schools, is particularly concerning, as it indicates the vulnerability of students in these environments. The rise in tertiary-level crimes, including rape and GBH, further underscores the need for enhanced safety measures across all educational institutions in South Africa. Addressing these trends requires urgent intervention, including improved security measures, law enforcement presence, and community engagement to create safer learning environments. The research presented in this article offers substantial contributions to Victimology, particularly in understanding the safety challenges faced by universities. For Criminologists, the data on crime rates, especially concerning theft, assault, and gender-based violence, provide a foundation for developing targeted crime prevention initiatives. For instance, Greeff et al. (2021) demonstrated that crime patterns in South African universities often mirror broader societal issues such as inequality and unemployment, which necessitate targeted interventions aimed at addressing these root causes within the campus setting [25].

Improving campus safety remains a critical priority for South African universities to create secure learning environments for all stakeholders. Addressing these challenges requires collaboration among university leadership, students, staff, and external stakeholders. Continuous assessment, policy refinement, and proactive measures are essential for improving health and safety on campuses. To address these systemic challenges, universities must adopt mandatory annual safety audits, establish survivor support centres, implement functional campus alert systems, and ensure enforcement of GBV protocols. National-level policy oversight and funding conditionality can incentivise compliance and improve institutional accountability. Addressing these failures is not only a national imperative but also contributes to advancing global development targets, particularly SDG 3 and SDG 4, by safeguarding health and promoting inclusive, secure access to higher education.

### 5.1. Recommendations for Improving Health and Safety in South African Universities

Given the data, more robust safety measures, stricter law enforcement, and community-driven intervention programs are necessary to address the rising crime rates in educational institutions. The increase in GBH, rape, and murders underscores the urgency for targeted policies to enhance security in these spaces, ensuring the safety and well-being of students and educators alike. Without immediate and sustained intervention, the current crime trajectory suggests that violence in educational institutions will continue to rise in the coming years, posing a severe threat to the learning environment in South Africa. Universities must improve prevention strategies, support services, and reporting mechanisms to address these issues. To address the institutional safety challenges identified in this study, we propose the following evidence-informed policy actions:

Standard procedures and integration of Safety into Organisational Culture: Current research reveals that many students are not aware of the standard procedures for reporting crimes and emergencies, leading to underreporting. Universities are complex ecosystems shaped by organisational culture and individual behaviours. According to research, an employee’s performance is greatly impacted by the occupational health and safety culture, which includes policies and procedures, employee involvement, workplace environment, and top management commitment. Safety should be ingrained in the university’s ethos [54].

Policies often fail when they exist in isolation from daily practices. Cultivating a safety-conscious culture requires leadership commitment and consistent messaging. Again, complex administrative processes can delay policy implementation. Streamlining procedures and ensuring efficient communication between departments is essential. Clear lines of responsibility and accountability are crucial to avoid bureaucratic bottlenecks. Policies alone are insufficient without proper dissemination and understanding. Staff, students, and contractors need regular training on safety protocols, emergency procedures, and risk mitigation. Awareness campaigns can bridge this gap by promoting a safety culture and encouraging proactive behaviour. Universities often have established practices that resist adaptation. Implementing new safety policies may face resistance from stakeholders who perceive them as disruptive or unnecessary; however, effective change management strategies are necessary to overcome this barrier.

Cultural norms, leadership practices, and employee attitudes influence safety outcomes. The “safety climate” within institutions plays a pivotal role, thus including staff and students in prioritising safety. A great institution’s safety culture includes student engagement and participation. Students play a central role in campus safety. Engaging students in safety committees, awareness campaigns, and feedback loops fosters a sense of ownership. Policies should encourage student participation and empower them to raise safety concerns.

Security improvement and collaboration: The article by Dlamini and Olanrewaju (2021) proposes a conceptual framework for proactive campus security, involving universities, the government, and students [24]. It suggests that institutions should mitigate external factors such as student protests, poverty, and gender-based violence through engagement, support, and zero-tolerance policies. Overall, the research highlights the need for a systematic approach to campus security that includes standard operating procedures, information sharing, and addressing external factors. Implementing these measures can enhance the safety and security of students and the university as a whole. Collaboration with local police departments is essential for crime prevention initiatives and coordinating responses. Universities should ensure that all incidents are investigated and reported to relevant authorities, and that accurate incident data are essential for prevention and improvement. Institutions should mitigate external factors such as student protests and gender-based violence through engagement, support, and zero-tolerance policies. Criminological research can also guide campus security enhancements, such as improved lighting, surveillance, and the presence of trained security personnel to deter criminal activity [10].

*Real-Time Campus Alert and Response Systems*: Institutions must implement real-time digital alert systems (e.g., SMS, app-based, or campus-wide public address systems) that allow students and staff to report emergencies instantly and receive critical safety updates. These systems should be integrated with campus security and local law enforcement to reduce response delays.

*Conditional Safety-Linked Funding Mechanisms:* The Department of Higher Education and Training (DHET) should introduce a conditional grant system wherein public funding is linked to the fulfilment of minimum campus safety benchmarks. Institutions that fail to meet standards for GBV response, security infrastructure, or mental health support should face escalated oversight and corrective mandates.

Mandatory Annual Safety Audits: Universities should be legally required to conduct and publish annual safety audits, including evaluations of physical infrastructure, security protocols, and emergency preparedness. These audits should be externally verified to ensure transparency and to support continuous improvement.

GBV Survivor Support Units: Dedicated survivor support offices staffed by trauma-informed professionals must be established at all public universities. These units should offer psychosocial care, legal advisory services, and facilitate access to justice through confidential reporting systems. Policies must ensure non-retaliation and survivor protection throughout.

Educational Programs: A holistic approach combining physical security measures, education, and community involvement contributes to a safer university environment. Safety education programs cover topics like personal safety, sexual assault prevention, and emergency preparedness [51,52,53]. In higher education institutions in South Africa, mental health programs have been severely lacking. Reports indicate that more than one-third of students experience mental distress during their university life, yet there is a lack of facilities and trained professionals to address this issue [51,52]. To effectively develop intervention policies and strategies for controlling student mental distress, it is crucial to understand its extent and predictors among students. This understanding will greatly assist practitioners and policymakers.

Including Criminologists and Victimologists: To ensure a comprehensive approach to campus safety, universities should actively engage Criminologists in the design and implementation of crime prevention strategies. Criminologists can assess risk factors, analyse crime trends, and provide data-driven recommendations for improving campus security measures. Research has shown that implementing surveillance systems and conducting regular safety audits, guided by criminological research, can significantly reduce incidents of theft, assault, and other crimes on university campuses [10]. Collaboration with Criminologists can also help universities stay ahead of emerging crime trends, particularly with the integration of smart technologies in campus security [22].

At the same time, Victimologists should play a central role in designing victim-centred support systems. Victimologists can work with university counselling services to create trauma-informed care models that assist victims of crime, especially those affected by gender-based violence. Research has shown that the availability of trauma counselling and legal support can dramatically improve the well-being of victims and encourage reporting [9]. Victimologists should also be involved in awareness campaigns that educate students about the importance of reporting crimes and seeking help when needed. In addition, their involvement in policy-making can ensure that victims’ rights are prioritised in university protocols and safety procedures [7].

Furthermore, interdisciplinary collaboration between Criminologists, Victimologists, and campus security personnel is essential. Universities should establish safety committees that include experts from these fields to provide ongoing guidance on security improvements and support initiatives for victims. This approach would ensure a balance between preventing crime and addressing the needs of victims, creating a safer and more supportive campus environment.

### 5.2. Limitations

This study is based entirely on secondary data, including media articles, South African university press releases, public reports, and academic publications. As such, it does not incorporate primary data from students, staff, or administrators, and therefore cannot fully capture the lived experiences or perceptions of those directly affected by campus safety risks. Therefore, the study acknowledges several limitations, including reliance on secondary data sources, potential underreporting of incidents, limited access to South African universities’ internal reports, and the challenge of separating university-specific data from broader educational institution statistics. The document selection process, while systematic, was constrained by the availability of sources that met the inclusion criteria. Furthermore, the reliance on publicly reported incidents may result in the underrepresentation of unreported or institutionally suppressed events. These limitations are addressed through the triangulation of multiple data sources and explicit acknowledgement of data constraints.

The thematic analysis, though conducted rigorously, is subject to interpretive limitations common in qualitative work. Coding and categorisation decisions may reflect underlying assumptions and the researcher’s positionality. Additionally, the lack of disaggregated national data for universities limited our ability to quantify institutional differences. Finally, while examples were drawn from a range of universities, the findings cannot be generalised to all institutions in South Africa. Nonetheless, they highlight systemic patterns that warrant policy attention and further empirical investigation.

## Figures and Tables

**Figure 1 ijerph-22-00989-f001:**
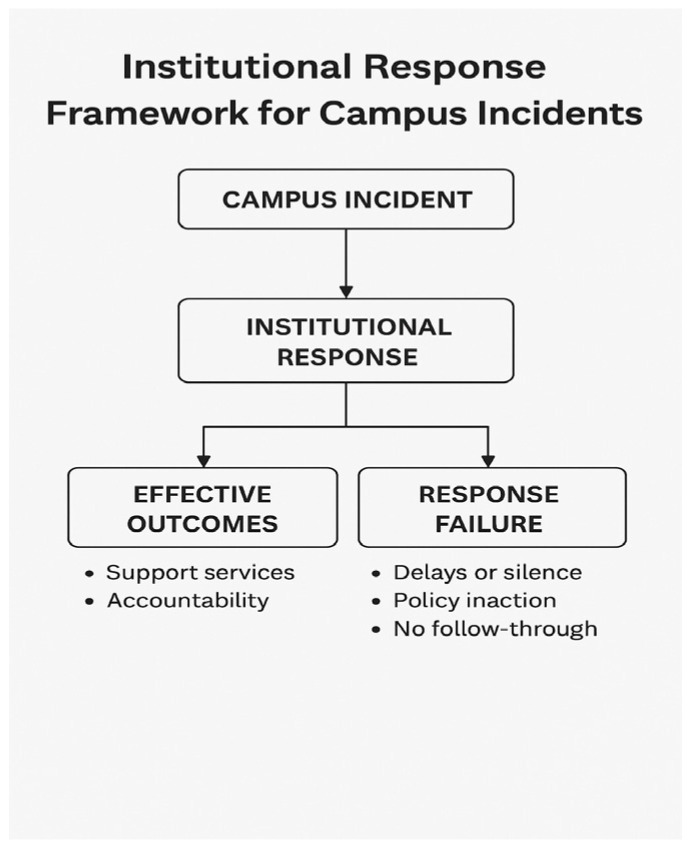
Institutional Response Framework for Campus Incidents.

**Table 1 ijerph-22-00989-t001:** Core elements of a public health framework for institutional safety.

Element	Description
Focus	Prevention of harm (injury, violence, disease) across populations
Level of Analysis	Structural, environmental, and systemic factors, not just individual acts
Assumptions	Most harm is preventable, not inevitable
Methods	Uses surveillance data, risk assessments, and root cause analysis
Intervention Strategy	Emphasises multi-level, multi-sectoral solutions (policy, design, education)
Example Issues	GBV, mental health, poor housing, school violence, substance abuse

## Data Availability

All data used in this study were derived from publicly available institutional documents, news reports, and official statements. A list of sources and document references is available upon reasonable request from the corresponding author. No new datasets were generated or proprietary data used during this study.

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
