# Peer review of "Health and Safety Challenges in South African Universities: A Qualitative Review of Campus Risks and Institutional Responses"

_ijerph, 2025, doi:10.3390/ijerph22070989_

Round 1
Reviewer 1 Report
Comments and Suggestions for Authors
The study addresses a critical issue concerning health and safety within South African universities, a topic of great significance given the current challenges faced in these environments. However, despite its relevance, the manuscript is not suitable for publication in its present form due to several significant shortcomings. Below, I outline my reasons for this assessment:
- Introduction
The study examined health and safety concerns at universities in South Africa. While the author provided background information, the aim and objectives of the study were not clearly stated. As a result, it is challenging to discern the specific goals in the introduction section of the manuscript. Although the manuscript addresses health and safety issues broadly, it lacks a clear focus.
- Material and methods
In this section, the authors addressed the study design, study population, sample size, data sources, data collection methods, and data analysis technique. However, the rationale behind the chosen research design was not clearly explained, and the criteria for selecting the study population and sample, as well as data sources and collection methods, were lacking. Additionally, some of the information in this section might be more appropriate if it were included in the introduction of the manuscript. For example, details from line 87 (“the focus area...”) to line 93 could fit much better in the introduction section of the manuscript.
- Results
The findings presented in the study extend beyond the intended focus on universities, encompassing a broad spectrum of educational institutions, including schools and daycare centres. This divergence from the original scope undermines the study’s purpose, which was specifically designed to concentrate on universities.
- Discussion
The discussion section does not clearly relate to the study's results. This section is meant to contextualise and analyse the study's findings; however, it lacks clarity and coherence, making it difficult to understand how the results are being interpreted.
- Contradictions
- In the abstract (line 14), the author indicates that the study is shaped by public health and victimology frameworks to explore systemic risk factors and institutional responses. However, on page 2, lines 64 to 65, it is stated that the article is shaped by criminology and victimology in tackling the ongoing issues of violence. This presents a contradiction: does the study rely on public health and victimology, or is it guided by criminology and victimology?
- On page 6, lines 232 to 233, the authors state, “This section delves into specific incidents that have taken place at South African universities over the past 10 years, starting from 2013”. However, this assertion appears to conflict with the information provided in the abstract (line 14), where it is clearly stated that the study will focus on the period from 2015 to 2024. This inconsistency raises concerns about the scope and time frame of the research being presented.
I have added comments to the original manuscript for the authors to take into account.

Author Response
Good day. Find attached

Reviewer 2 Report
Comments and Suggestions for Authors
Revise the methodological description. This is not a mixed-methods study. Accurately define it as a document analysis with a qualitative approach. Clearly describe the selection criteria, sources, and data analysis procedures.
Add a detailed explanation of how documents were selected, how many were included, and the criteria for inclusion and exclusion. Clarify how the data was coded and categorized for analysis.
Improve the structure of the results by organizing them into thematic categories.
Expand the limitations section, explicitly acknowledging the absence of primary data, reliance on secondary sources, and potential impacts on the validity and generalizability of the findings.
I don't feel qualified to make recommendations of this nature.
Author Response
Good Day
Find attached

Reviewer 3 Report
Comments and Suggestions for Authors
Thank you for the opportunity to review this article. I was very interested in the title as it is an important and timely topic. However, the article needs significant development before it is publishable. I encourage the author/s to continue to work on the article as it will potentially make a valuable contribution to the existing literature in the future.
One of the most critical improvements that is needed is for the article's thesis to be strengthened. The title of the article is very promising but it often goes off track. For example, the focus needs to be firmly on universities - not 'educational institutions'. I encourage the authors to re-engage with the key message they want to get across and then really focus in on it by ensuring the structure is logical.
Another very important point is that the article is currently under-researched and referenced for a peer-reviewed international level paper. At this level the significant body of available literature (internationally) needs to inform the analysis and should be synthesised throughout the paper. South Africa is not the only country in which these issues are important and difficult.
The article needs a clear introduction in which its thesis is clearly set out, the structure of the article is signposted, and the conclusion is foreshadowed.
The methodology needs significant development for clarity and rigour.
The presentation and discussion of the results needs to be made clearer and tightened in terms of having a logical flow.
The article also needs a very thorough proof-read for accuracy and clarity of expression. For example, the under the heading 'Results' the template instructions - which should have been deleted - have been left in. And on page 9 there is a heading numbered 4.2 that comes after heading 4.3 on page 8. In terms of citing legislation etc the South African jurisdiction needs to be made clear. There is a lot of over-capitalisation (particularly of Criminologists and Victimologists) and repetition that need to be addressed.
Author Response
Good Day
Find attached

Round 2
Reviewer 1 Report
Comments and Suggestions for Authors
The manuscript has shown improvements compared to the initial submission. However, some sections need further revision to meet publication standards. Comments and recommendations have been included directly in the manuscript to guide the authors in making the necessary adjustments.

Author Response
Good Day
See attached response report.
Regards

Reviewer 2 Report
Comments and Suggestions for Authors
The authors have addressed all previous recommendations or comments, further improving the manuscript.
Author Response
Comment 1: The authors have addressed all previous recommendations or comments, further improving the manuscript.
Respond 1: Thank you